# Circulating microRNAs as Biomarkers of Various Forms of Epilepsy

**DOI:** 10.3390/medsci13010007

**Published:** 2025-01-08

**Authors:** Elena E. Timechko, Kristina D. Lysova, Alexey M. Yakimov, Anastasia I. Paramonova, Anastasia A. Vasilieva, Elena A. Kantimirova, Anna A. Usoltseva, Albina V. Yakunina, Irirna G. Areshkina, Diana V. Dmitrenko

**Affiliations:** 1Department of Medical Genetics, Clinical Neurophysiology of Postgraduate Education, V.F. Voyno-Yasenetsky Krasnoyarsk State Medical University, Russian National Research, Krasnoyarsk 660022, Russiakris_995@mail.ru (K.D.L.); korolek_xd@bk.ru (A.I.P.); drroptimusprime@gmail.com (A.A.V.); a.usoltseva@list.ru (A.A.U.); strotskaya1992@mail.ru (I.G.A.); 2Federal State Budgetary Educational Institution, Samara State Medical University, Ministry of Healthcare of the Russian, Samara 443079, Russia

**Keywords:** temporal lobe epilepsy, juvenile myoclonic epilepsy, microRNA, drug-resistance, biomarkers, expression profile

## Abstract

**Background**: Epilepsy is a group of disorders characterized by a cluster of clinical and EEG signs leading to the formation of abnormal synchronous excitation of neurons in the brain. It is one of the most common neurological disorders worldwide; and is characterized by aberrant expression patterns; both at the level of matrix transcripts and at the level of regulatory RNA sequences. Aberrant expression of a number of microRNAs can mark a particular epileptic syndrome; which will improve the quality of differential diagnosis. **Materials and Methods**: In this work; the expression profile of six microRNAs was analyzed: hsa-miR-106b-5p; hsa-miR-134-5p; hsa-miR-122-5p; hsa-miR-132-3p; hsa-miR-155-5p; and hsa-miR-206-5p in the blood plasma of patients suffering from temporal lobe epilepsy (n = 52) and juvenile myoclonic epilepsy (n = 42); n—amount of participants; in comparison with healthy volunteers. The expression analysis was carried out using RT-PCR. Mathematical processing of the data was carried out according to the Livak method. **Results**: A statistically significant change in the expression of hsa-miR-106b-5p; hsa-miR-134-5p; hsa-miR-122-5p; and hsa-miR-132-3p was found. An increase in the expression of hsa-miR-134-5p and hsa-miR-122-5p was registered in the group of patients with temporal lobe epilepsy compared to the control; as well as an increase in the expression of hsa-miR-132-3p and hsa-miR-106b-5p in the juvenile myoclonic epilepsy group compared to the control. hsa-miR-122-5p; 106b-5p; 132-3p are also able to discriminate groups with different syndromes. Additionally; a number of microRNAs are able to discriminate patients with drug-resistant and drug-sensitive forms of epilepsy from the control; as well as patients with hippocampal sclerosis and patients without hippocampal sclerosis from the control. Conclusion. Our data allow us to propose these microRNAs as plasma biomarkers of various epileptic syndromes

## 1. Introduction

Epilepsy syndrome is a disorder or group of disorders that, according to the definition of the International League Against Epilepsy (ILAE), is characterized by a cluster of clinical and EEG features, often supported by specific etiologic data (structural, genetic, metabolic, immune, and infectious) [1]. According to the World Health Organization (WHO), epilepsy affects approximately 50 million people worldwide [2]. Epileptic syndromes are most often classified based on the age of onset of the disease, as well as the type of seizures, in particular, the following: focal; generalized; and focal and/or generalized [3]. The diagnosis of a specific epileptic syndrome is based on anamnestic data, EEG results, neuroimaging, and genetic data; however, in a number of cases, the presence of generalized spike-wave discharges was detected in individuals not suffering from an epileptic syndrome [4,5]. Also, in a critical review, Seneviratne U. et al. (2014) [6] found that in a number of cases, idiopathic generalized epilepsies (IGE) are accompanied by a number of focal abnormalities. Also [7], cases of hippocampal sclerosis have been reported in patients with IGE. The situation is complicated by the fact that, in particular, IGE can make its debut in adulthood [8]. Thus, cases of misdiagnosis of various epileptic syndromes are not uncommon, which is why there is a great need to identify additional markers of the disease that can complement and facilitate the diagnosis of epileptic syndromes.

Biomarkers are biochemical indicators that can detect changes or potential changes in the structure and function of cells and subcellular structures of systemic organs and tissues [9]. The most attractive approach is to search for circulating biomarkers, due to the ease and speed of obtaining biological material for analysis [10]. The usage of circulating molecular markers in epilepsy is justified, since it is known that seizure activity is accompanied by impaired permeability and dysfunction of the blood–brain barrier [11,12], which allows us to study tissue leak markers in biological fluids. To date, a significant number of circulating markers of epilepsy have been proposed. In particular, among the following protein molecules: HMGB1 [13,14], BDNF [15,16], DAG [17], NSE and S100 [18,19] and others, more information on protein markers can be found in the work of Banote R.K (2022) [20]. Analysis of aberrant gene expression and identification of circulating epilepsy biomarkers based on it is complicated by the difficulty in conducting mRNA transcriptomic experiments in biological fluids. MicroRNAs, short non-coding RNAs (20–24 bp) that function through sequence-specific binding to the 3′ untranslated regions (UTRs) of target mRNAs, are increasingly being considered as potential epilepsy biomarkers [21]. It is known that many microRNAs are specifically expressed in different cells of the brain [22], making them good candidates for epilepsy biomarkers.

A significant amount of data has been published describing microRNA expression patterns in epilepsy, both in models and in various patient biopsies. For example, increased expression of miR-27a-3p has been observed in the hippocampus of model rats [23], miR-146a in chronic TLE, and in the brain of a rat model [24]. A detailed review of microRNAs and their role in epilepsy is presented in the work of Wang J. et al. (2021) [25].

The existing data providing information on microRNA aberrations in epilepsy remain rather fragmented, and there is a need for new data. For this study, we selected six microRNAs for analysis: hsa-miR-134-5p, hsa-miR-106b-5p, hsa-miR-122-5p, hsa-miR-132-3p, hsa-miR-155-5p, and hsa-miR-206, whose expression was studied in two epileptic syndromes: TLE and JME in comparison with the controls, and blood plasma was studied.

## 2. Materials and Methods

### 2.1. Ethical Considerations and Patient Recruitment

The study was conducted in accordance with the recommendations of the Declaration of Helsinki [26] and approved by the Local Ethics Committee of the Krasnoyarsk State Medical University named after Professor V.F. Voyno-Yasenetsky, protocol number: 102/2020 dated 27 November 2020.

To investigate plasma miRNA expression, patients diagnosed with TLE (n = 52) and JME (n = 42) were consecutively recruited into experimental groups according to ILAE criteria. Diagnoses were made by an expert in epileptic neurological disorders and neuroimaging.

The following tests were performed on the patients of the experimental groups: HADS and NHS3. The main characteristics of the patients of the two groups are presented in Table 1.

Inclusion criteria is as follows:
-Confirmed diagnosis of TLE or JME;-Age 18–55 years;-Signed informed consent;-Absence of signs of infectious disease during the sampling of biomaterial.

The control group in the study included healthy volunteers, matched by age and gender, without neurological diseases and signs of infectious diseases, as well as without chronic somatic pathologies in the stage of decompensation.

The anamnesis collection included the following: study of the neurological status, assessment of the frequency and severity of epileptic seizures (Hospital Anxiety and Depression Scale (HADS), National Hospital Scale (NHS-3)), analysis of ongoing antiepileptic therapy, response to AEDs, EEG video monitoring, and MRI of the brain. All patients and healthy volunteers included in the study were of Slavic origin, born and living in the Siberian region of the Russian Federation.

Also, all patients recruited to the study groups had different durations of the disease. The difference between the TLE and JME groups was not significant; however, we indicate this fact as one of the limitations of our study.

All patients with epilepsy took antiepileptic drugs, most often including Valproic acid, Levetiracetam, Lamotrigine, Oxcarbazepine and Lacosamide. We cannot exclude the factor of taking AEDs as a potential inducer of aberrant expression of the studied microRNAs, therefore we consider this fact as one of the limitations of the study.

Whole blood from the cubital vein was collected in EDTA vacutainers and then centrifuged to separate the plasma fraction according to standard protocols [26]. Blood for subsequent analysis was taken at a point that corresponded to the duration of the disease. The separated plasma was stored in a low-temperature freezer at −80 °C until further analysis.

### 2.2. RNA Isolation and RT-PCR

Total RNA was isolated from blood plasma using the RIBO-sorb kit (Helikon, Moscow, Russia, article number: K2-1-Et-100) according to the manufacturer’s protocol. Total RNA (1 μg) was subjected to reverse transcription using the MMLV-RT kit (Eurogen, Moscow, Russia, article number: SK021) according to the manufacturer’s protocol and specific stem-loop primers.

Real-time PCR was performed using a Rotor-Gene Q 2plex Priority Package Plus thermal cycler (QIAGEN; Germantown Road, Germantown, MD, USA) and a commercial real-time PCR kit containing 2.5 PCR mix (Synthol; Moscow, Russia, article number: M-428) supplemented with specific F- and R-primers (0.9 μL, 10 pM) and a fluorescent probe (0.5 μL, 10 pM). All samples were analyzed in triplicate. Primers and probes for reverse transcription and PCR were designed using the srnaprimerdb software (http://www.srnaprimerdb.com; accessed on 20 November 2024). Cycle threshold (Ct) parameters obtained after PCR-RA that exceeded the 40th cycle were excluded from further analysis.

MicroRNA expression level was calculated by the Livak method [27].

The selection of microRNAs for subsequent analysis was based on literature data. A search was conducted for publications with the following keywords: microRNA and epilepsy, microRNA and neuroinflammation, microRNA and excitotoxicity, microRNA and neurodegeneration, microRNA and gliosis. The search was carried out in the following aggregators: Pubmed, ScienceDirect, Google Scholar.

The expression of the following miRNAs was analyzed: hsa-miR-134-5p, hsa-miR-106b-5p, hsa-miR-122-5p, hsa-miR-132-3p, hsa-miR-155-5p, and hsa-miR-206-5p. Hsa-mir-191 was used as reference gene due to its stability in plasma samples [28]. Additionally, this microRNA has already been used for normalization in epilepsy studies [29].

### 2.3. Data Analysis

The distribution analysis of miRNA expression data in groups was performed using the Shapiro–Wilk statistical test. The median and 25–75 percentiles (Me [LQ; UQ]) were used to describe the amount of data with abnormal accumulation. To compare several groups on a quantitative basis, non-parametric analysis of variance (the Kruskal–Wallis test) was used with Benjamani–Hochberg *p* value correction, The Mann–Whitney test with Benjamani–Hochberg *p* value correction has been carried out to compare the two groups.

Spearman’s correlation coefficient (r) was used to assess the relationship between quantitative traits with non-normal distribution. Intergroup differences were recognized as statistically significant at *p* < 0.05.

To assess the quality of the classification, ROC analysis with the determination of the area under the curve (AUC), as well as constructing a Random Forest model and decision tree model, were used.

Data analysis of the miRNAs expression and data visualization were performed using Python and the following libraries: pandas, numpy, scipy, plotly, seaborn, sklearn, upsetplot, and grapher. GO term enrichment was conducted with the following datasets: GO BP, KEGG, and GO MF.

## 3. Results

Six selected miRNAs—hsa-miR-134-5p, hsa-miR-106b-5p, hsa-miR-122-5p, hsa-miR-132-3p, hsa-miR-155-5p, and hsa-miR-206—were analyzed for tissue distribution. According to microRNA Tissue Atlas (https://ccb-web.cs.uni-saarland.de/tissueatlas2, accessed on 20 November 2024), selected microRNAs were highly enriched in many brain regions (Figure 1). In particular, these microRNAs were detected in the following: the brain, all lobes of the brain, the hippocampus, white matter, etc. Enrichment of the brain with the studied microRNAs allows for further analysis.

The EpiMirBase database (https://www.epimirbase.eu, accessed on 20 November 2024) also demonstrated the association of selected microRNAs with epilepsy.

We evaluated the expression of selected miRNAs in the blood plasma of different groups. Distribution tests reveal that the biggest amount of data have an abnormal distribution pattern (Figure 2), so we conducted the Kruskal–Wallis test to compare these groups.

The analysis revealed statistically significant changes (*p* < 0.05) in the expression of the following miRNAs in the TLE group compared to the control: log10FC; hsa-miR-134-5p = 1.12; hsa-miR-106b-5p = 0.23; hsa-miR-122-5p = 1.37; and hsa-miR-132-5p = 0.25. In the JME group, the following data were obtained compared to the control: hsa-miR-134-5p = 0.92; hsa-miR-106b-5p = 0.75; hsa-miR-122-5p = 0.96; and hsa-miR-132-5p = 0.69. hsa-miR-155-5p and hsa-miR-206-5p did not show statistically significant changes in expression pattern (Figure 3).

Post hoc analysis between the groups revealed the following:

Expression of hsa-miR-134-5p microRNA significantly differentiates TLE from the control (l0f = 1.12, *p* = 0.002); hsa-miR-106b-5p allows discrimination of TLE from JME (l10f =−0.52, *p* = 0.023) and JME from the control (l10f = 0.75, *p* = 0.003); hsa-miR-122-5p differs in TLE groups compared to the control (l10f = 1.37, *p* = 0.00002) and JME compared to the control (l10f = 0.96, *p* = 0.020) and also TLE compared to JME (l10f = 0.41, *p* = 0.031); and hsa-miR-132-3p differs in TLE groups compared to JME (l10f = −0.44, *p* = 0.044) and JME compared to the control (l10f = 0.69, *p* = 0.004). Plots describing significantly changed expressions between TLE and JME are exhibited in Figure 4.

Both the TLE group and JME group showed statistically significant increase in plasma expression for microRNAs: hsa-miR-134-5p; hsa-miR-106b-5p; hsa-miR-122-5p; and hsa-miR-132-5p in comparison with the control.

Within the group of patients with TLE, plasma expression of miRNAs was compared in MRI-positive (presence of hippocampal sclerosis, n = 25) patients and MRI-negative (absence of hippocampal sclerosis, n = 27) patients, as well as in the controls. The analysis demonstrated statistically significant differences between the MRI-positive groups and the controls, and MRI-negative groups and the controls. No statistically significant differences were found between the MRI-positive and MRI-negative groups (Figure 5).

Also, using machine learning algorithms (the training dataset includes classic 0.2 of the full one), a Random Forest classification model was created, as well as identifying factors that have the greatest impact on classification. The most significant factor for classifying patients as MRI-positive and MRI-negative was the expression of hsa-miR-206-5p, but with a low accuracy value of 0.25. The partial dependence graph demonstrates an increase in the expression of hsa-miR-206 for a group of patients with hippocampal sclerosis (Figure 6).

In addition, we constructed a decision tree, a nonparametric supervised learning algorithm used for classification problems with depth = 7 (Figure 7). The Gini index, specified for each node, measures impurity in decision nodes, helping to create efficient partitions. The Gini index ranges from 0 to 1, where «0» indicates that all elements belong to a particular class or there is only one class (pure), and «1» indicates that elements are randomly distributed across different classes (impure).

Additionally, within the TLE group, a comparison of expression patterns was performed between drug-resistant (DR, n = 18) and drug-sensitive (DS, n = 34) patients. hsa-miR-134-5p and hsa-miR-122-5p successfully discriminated DR patients (l10f = 0.97, *p* = 0.031; l10f = 1.22, *p* = 0.005 for hsa-miR-134-5p and hsa-miR-122-5p, respectively) and DS patients from the control (l10f = 1.20, *p* = 0.004; l10f = 1.46, *p* = 0.00009 for hsa-miR-134-5p and hsa-miR-122-5p, respectively) (Figure 8). Also hsa-miR-155-5p can discriminate DS patients from the controls (l10fc = 0.35, *p* = 0.03) and DR patients from DS patients (l10fc = 0.15, *p* = 0.04).

A Random Forest and decision tree model were also created to classify the data and identify the factors that have the greatest impact on classification (Figure 9). The most significant factor for classifying drug-sensitive and drug-resistant patients was the expression of hsa-miR-155-5p with an accuracy of 0.73. The partial dependence plot demonstrates a characteristic decrease in hsa-miR-155-5p expression for the DR group of patients. L10fc for DR patients in comparison with DS patients was −0.499 but the *p* value was near 0.05. The studied samples were of different sizes, which is a limitation, therefore the presence or absence of statistically significant differences may be a consequence of this limitation.

Interestingly, the obtained data are confirmed by the conducted ROC analysis, classifying drug-resistant and drug-sensitive patients with TLE based on the expression data of hsa-miR-155-5p microRNA, in particular, AUC = 0.67 with CI [0.49; 0.79] (Figure 10); however, an attempt to discriminate these groups by other microRNAs does not allow this to be achieved: hsa-miR-134-5p: AUC = 0.53, CI [0.30; 0.64], hsa-miR-106b-5p: AUC = 0.50, CI [0.34; 0.66], hsa-miR-122-5p: AUC = 0.55, CI [0.28; 0.61], hsa-miR-132-3p: AUC = 0.54, CI [0.26; 0.62], hsa-miR-206-5p: AUC = 0.51, CI [0.35; 0.67].

The presence of statistically significant differences was also assessed for all other anamnestic characteristics. Among the significantly different ones, the following were identified: the expression pattern of hsa-miR-206 between the groups of compensated and uncompensated disease course in the JME cohort, as well as for hsa-miR-132-3p between different frequencies of bilateral tonic–clonic seizures (BTCS) in the TLE cohort (Figure 11 and Figure 12). No statistically significant differences in the expression of the studied microRNAs for gender, the presence or absence of serial course, or the frequency of GTCS were found. All the studied data are compiled in Table 2.

For each group, a binary classification analysis was performed using ROC curves, using expression data of only four statistically significant miRNAs. Classification of TLE from the control demonstrated hsa-miR-134-5p AUC (area under the curve) = 0.69, CI (confidence interval) [0.59; 0.80]; hsa-miR-122-5p AUC = 0.75, CI [0.67; 0.86]; the fusion of all four miRNAs demonstrated AUC = 0.79. hsa-miR-106b-5p and hsa-miR-132-3p with AUC = 0.5 and CI [0.39; 0.63]; AUC = 0.56 and CI [0.45; 0.69], respectively, do not allow the discrimination of TLE from the control with sufficient reliability.

Classification of JME from the control demonstrated the following: hsa-miR-106-5p AUC = 0.69, CI [0.56; 0.79]; hsa-miR-122-5p AUC = 0.65, CI [0.53; 0.77]; hsa-miR-132-3p AUC = 0.68, CI [0.57; 0.79], and the fusion of all four microRNAs demonstrated AUC = 0.71. hsa-miR-134-5p with AUC = 0.59 and CI [0.46; 0.72] does not allow the discrimination of JME from the control with sufficient reliability.

All obtained ROC curves are presented in Figure 13.

Also, additional ROC analysis was performed for TLE and JME groups based on the expression data of significantly different microRNAs: hsa-miR-106b-5p, hsa-miR-122-5p, and has-miR-132-3p. hsa-miR-106b-5p allows differentiating TLE versus JME: AUC = 0.64, CI [0.26; 0.49]. hsa-miR-122-5p allows differentiating TLE versus JME: AUC = 0.58, CI [0.48; 0.72]. hsa-miR-132-3p allows differentiating TLE versus JME: AUC = 0.62, CI [0.27; 0.49]. The fusion of both microRNAs demonstrated AUC = 0.71 (Figure 14). Additionally, we performed ROC analysis for the remaining microRNAs: hsa-miR-134-5p: AUC = 0.55, CI [0.43; 0.67]; hsa-miR-155-5p: AUC = 0.51, CI [0.38; 0.61]; hsa-miR-206-5p: AUC = 0.57, CI [0.32; 0.55].

Analysis of data correlations demonstrated the presence of statistically significant correlations between the expression of various microRNAs. However, statistically significant correlations between microRNA expression and clinical characteristics of patients, such as age of onset, duration of the disease, and test scores, were not revealed (Figure 15).

Spearman correlation analysis demonstrated the presence of significant but not very strong correlations between the expression of microRNAs: hsa-miR-134-5p and hsa-miR-106b: r = 0.53; hsa-miR-134-5p and hsa-miR-132-3p: r = 0.36; hsa-miR-106b-5p and hsa-miR-122-5p: r = 0.48; hsa-miR-106b-5p and hsa-miR-132-3p: r = 0.37; hsa-miR-122-5p and hsa-miR-155-5p: r = 0.22; hsa-miR-122-5p and hsa-miR-132-3p: r = 0.38; hsa-miR-206 and hsa-miR-132-3p: r = 0.40 and one strong correlation between hsa-miR-134-5p and hsa-miR-122-5p: r = 0.72.

Comparison of groups by disease duration, age at the time of the study, gender, seizure frequency, and microRNA expression did not reveal statistically significant differences.

For each of the studied miRNAs, gene targets obtained from the miRtaRBase database with three strong validations (qPCR, Reporter assay, Western blot) were examined. An analysis of the presence of intersections between the gene targets of these miRNAs was performed, which demonstrated the presence of intersections (Figure 16).

Analysis of biological processes regulated by gene targets revealed an interesting cluster of genes involved in CNS processes (Figure 17).

A significant pool of gene targets under the control of the studied microRNAs is involved in the process of apoptosis, neural projection, oxidative stress, and glial proliferation.

## 4. Discussion

In this paper, we present the analysis of the plasma expression of six microRNAs. The analysis revealed four microRNAs whose expressions are significantly dysregulated. In particular, the expression of hsa-miR-134-5p microRNA significantly distinguishes TLE from the controls. hsa-miR-106b-5p allows the discrimination of TLE from JME and JME from the controls. hsa-miR-122-5p differs in TLE compared to the controls and JME compared to the controls. hsa-miR-132-3p differs in TLE compared to JME and JME compared to the controls. The data obtained during the analysis of plasma expression allows us not only to distinguish pathology from the controls, but also to discriminate between different epileptic syndromes, which can serve as a useful tool for additional diagnostics. In addition, blood plasma has an advantage over other biopsies due to its availability and ease of collection.

Hsa-miR-134-5p was significantly upregulated in the plasma of patients with TLE. This observation is confirmed with previous data. In particular, hsa-miR-134-5p was upregulated in the hippocampal tissues of children with mTLE [30], as well as in the brain tissues of animal models [31]. The antiseizure effect of inhibition of this microRNA has been demonstrated in a significant number of studies [32,33]. hsa-miR-134-5p is suggested to be involved in the control of dendritic spine morphology through targeting the LIM kinase-1 gene [34], which in turn inactivates cofilin through its phosphorylation. Repression of LIM kinase-1 by hsa-miR-134-5p results in abnormal dendritic spine architecture. Also, hsa-miR-134-5p was shown to be associated with the induction of expression of the pro-apoptotic transcription factor C/EBP homologous protein (CHOP) [35], while the silencing of hsa-miR-134-5p reduces *CHOP* and *Bim* expression in the hippocampus. Levels of this miRNA in the plasma of patients with mTLE were also increased in the study by Leontariti et al. [36]. Post hoc analysis of differences did not reveal statistical significance of this microRNA for the JME group, which may be explained by its specificity for the hippocampus [37]. Interestingly, despite the lack of statistically significant differences between the DS and DR patients, expression of this microRNA was downregulated in the DR group in comparison with DS, which contradicts some previously obtained data [38] and confirmed other studies [29]. This fact can be explained by the small sample of DR patients, as well as the different sample sizes; continuation of research in this area and an increase in the size of the studied samples can shed light on the observations obtained. Thus, our findings suggest that hsa-miR-134-5p may be a potentially good diagnostic plasma marker of TLE, and also, due to its ability to discriminate drug resistance and drug sensitivity from the controls, a potential prognostic marker of the disease course.

hsa-miR-106b-5p was significantly elevated in the plasma of JME patients compared with TLE, and the controls, which was also elevated in other studies, including the study by Wang J. et al. (2015) [38]. This microRNA was elevated in the plasma of patients with epilepsy, most of whom were patients with IGE. Also, the obtained data are confirmed by previously conducted studies [39,40], examining the expression of this microRNA in serum and blood plasma. In addition, hsa-miR-106b-5p was elevated in the IGE patient group [41]. It was revealed that hsa-miR-106b-5p has a proepileptogenic function due to the induction of neuroinflammatory processes through its effect on *TGFB* [42]. Hsa-miR-106b-5p is associated with apoptosis-associated caspase-3 and caspase-9 [43].

hsa-miR-122-5p has not been previously studied in the context of epilepsy. However, it has been associated with acute cerebral events such as stroke [44,45], as well as Alzheimer’s disease [46]. There is evidence that hsa-miR-122-5p is associated with the *FOXO3* gene and exerts a neuroprotective effect through the induction of the NF-κB cascade [47], which may be a compensatory response to seizure activity. In vitro data demonstrated that hsa-miR-122-5p was able to upregulate inflammatory factors and lead to microglial activation via targeting *MLLT1* and inhibiting the PI3K/AKT pathway [48]. If we take into account the hypothesis of neuroinflammation induction, it is known that neuroinflammation is not a process specific to a particular epileptic syndrome [49], but is, in general, a process that characterizes epileptogenesis; therefore, significantly increased expression of this microRNA can be characteristic of both TLE and JME. Nevertheless, our data on the expression of hsa-miR-122-5p allow us not only to discriminate the pathological phenotype from the control, but also to demarcate between different forms of epilepsy.

hsa-miR-132-3p has been shown to be overexpressed in the plasma of patients with JME. This miRNA has been most frequently studied in the context of TLE [50,51]. In the work of Tak A.Y. et al. (2024) [52], his microRNA was proposed as a diagnostic biomarker for IGE. The role of hsa-miR-132-3p as an inducer of inflammation is confirmed [53]. Also, hsa-miR-132-3p exerts proepileptic effects via modulation of the BDNF/TrkB pathway [54] and promotes the epileptogenic process. In the work of Martins-Ferreira R. et al. (2020), hsa-miR-132-3p was proposed as one of the diagnostic biomarkers of genetic generalized epilepsies (GGE) [55].

hsa-miR-155-5p and hsa-miR-206 showed an increase and decrease, respectively (for TLE), in expression in the experimental groups; however, this change in the expression profile was not statistically significant, despite the fact that these miRNAs have previously been reported to be associated with epilepsy [56,57]. hsa-miR-155-5p was also downregulated in DR patients in comparison with DS patients, but it could not discriminate DR and the control. Some studies discovered the fact that hsa-miR-155-5p is associated with a drug-resistant course of epilepsy [58,59], but in previous studies its expression was upregulated. Such results may be related to the characteristics of the sample and its relatively small size, which is a limitation of our study.

We conducted a study on the plasma expression of microRNAs using two experimental groups. The obtained data allow us to propose four of the studied microRNAs as potential plasma biomarkers of epileptic syndrome. In addition, the obtained data also allow us to successfully discriminate patients by their response to therapy, as well as the presence of hippocampal sclerosis.

## 5. Conclusions

The plasma expression of six microRNAs was analyzed: hsa-miR-134-5p, hsa-miR-106b-5p, hsa-miR-122-5p, hsa-miR-132-3p, hsa-miR-155-5p, and hsa-miR-206. A statistically significant increase in hsa-miR-134-5p and hsa-miR-122-5p expression were found in the plasma of patients with TLE compared to the control; this microRNA also successfully discriminates drug-resistant TLE and drug-sensitive TLE from the control, and hsa-miR-155-5p allows us to classify patients in drug resistance and drug sensitivity. A statistically significant increase in the expression of hsa-miR-106b-5p and hsa-miR-132-3p was also found in the plasma of patients with JME compared to the control. hsa-miR-122-5p was upregulated in patients with TLE in comparison with JME, while hsa-miR-106-5p and hsa-miR-132-3p were upregulated in patients with JME in comparison with TLE; hsa-miR-122-5p also allows discrimination between patients with and without hippocampal sclerosis from the control. The obtained results allow us to propose these microRNAs as plasma biomarkers of various epileptic syndromes. Our results are limited because only two epileptic syndromes were studied, while there are a significant number of them. Further in-depth research on this topic, expansion of the sample size, study of other types of epileptic syndromes, and other markers will allow us to better understand the role of microRNA in epileptic syndromes.

## Figures and Tables

**Figure 1 medsci-13-00007-f001:**
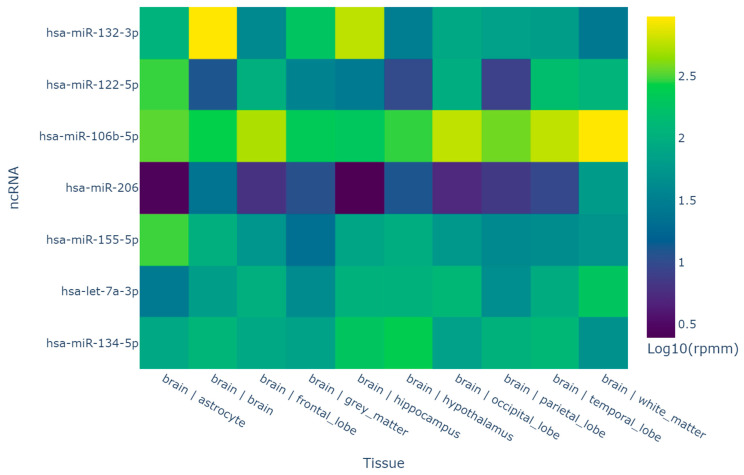
Expression of the studied microRNAs in the brain (rppm—reads per million normalization).

**Figure 2 medsci-13-00007-f002:**
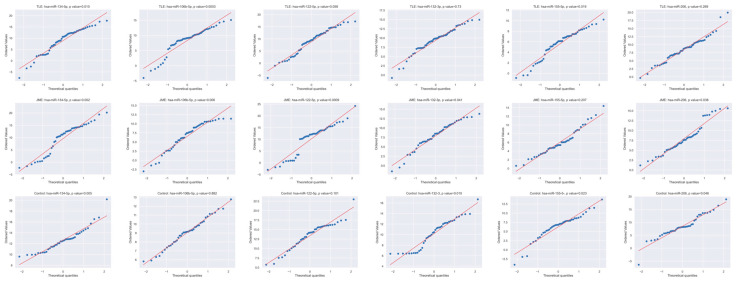
MicroRNA expression pattern distribution for TLE, JME, and control.

**Figure 3 medsci-13-00007-f003:**
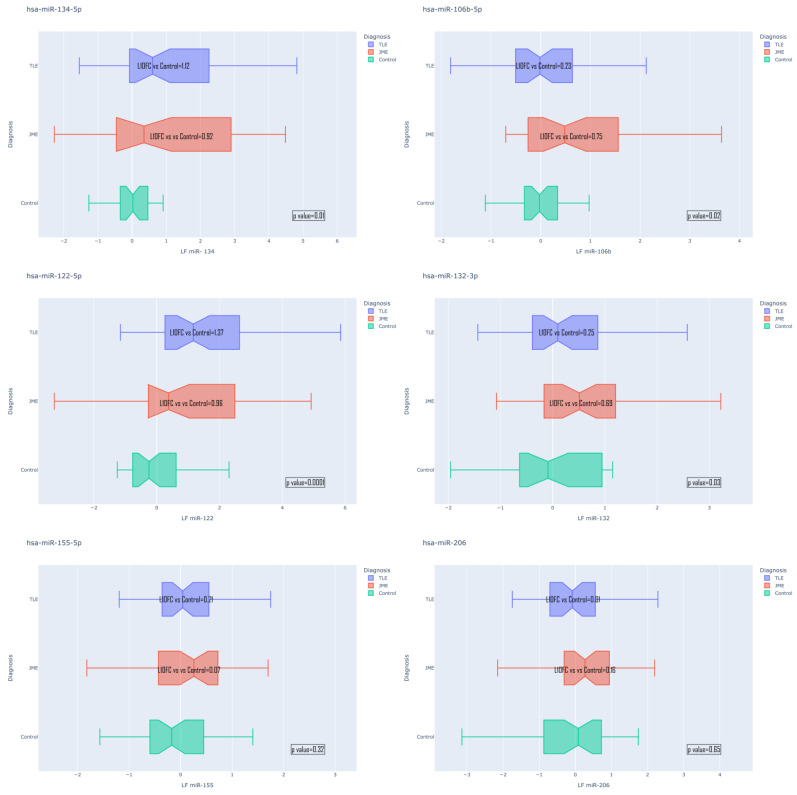
Differential MicroRNA expression patterns.

**Figure 4 medsci-13-00007-f004:**
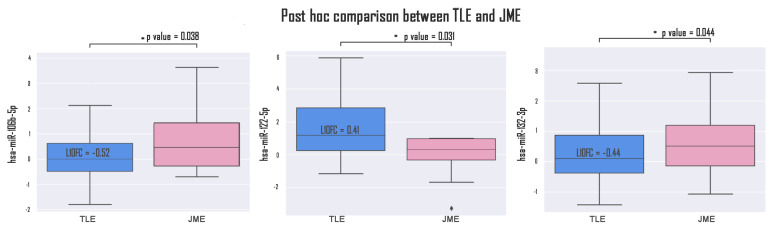
miR-106b-5p, miR-122-5p, and miR-132-3p expression in TLE and JME. *—*p* value < 0.05.

**Figure 5 medsci-13-00007-f005:**
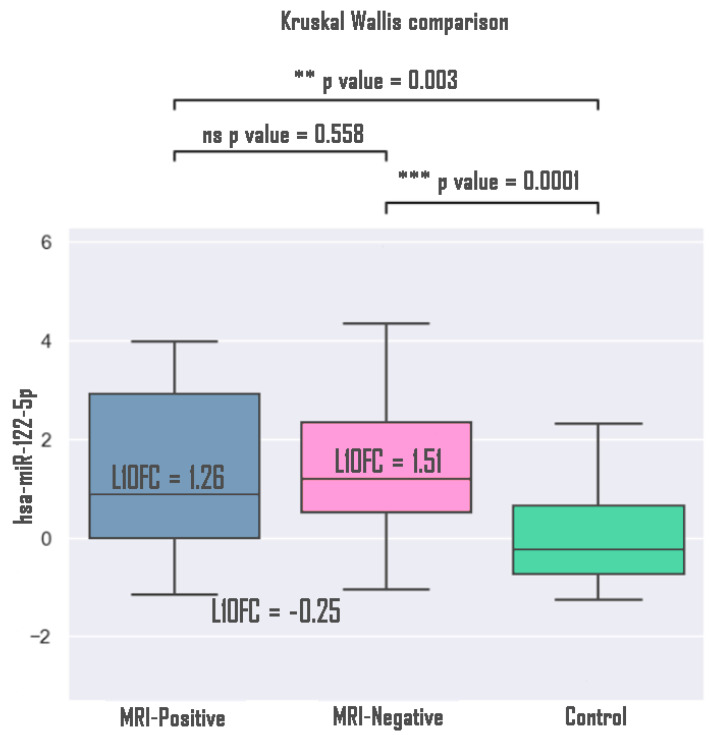
Expression of hsa-miR-122-5p microRNA in MRI-positive and MRI-negative patients compared to controls. **—*p* value < 0.01, ***—*p* value < 0.001.

**Figure 6 medsci-13-00007-f006:**
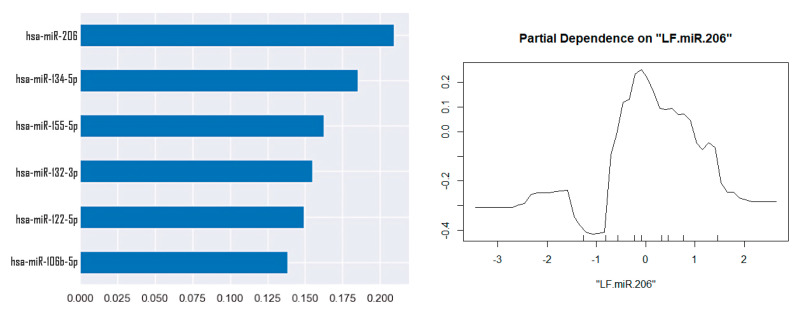
Random Forest for classification of MRI-positive and MRI-negative.

**Figure 7 medsci-13-00007-f007:**
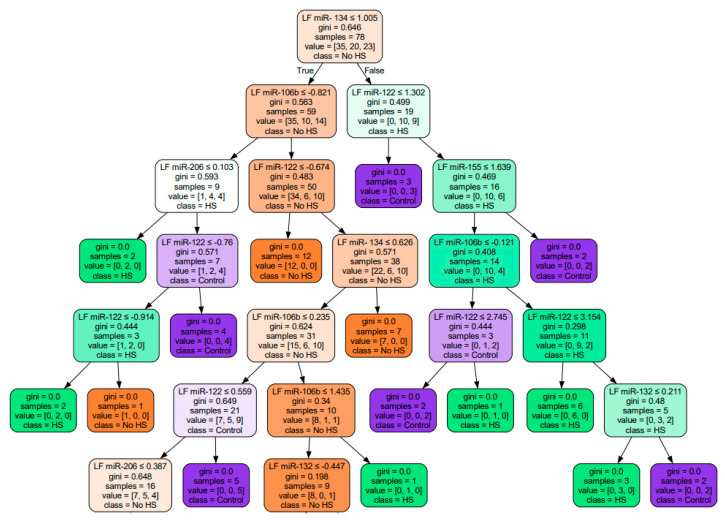
Decision tree for classification of MRI-positive, MRI-negative and control (gini-Gini index).

**Figure 8 medsci-13-00007-f008:**
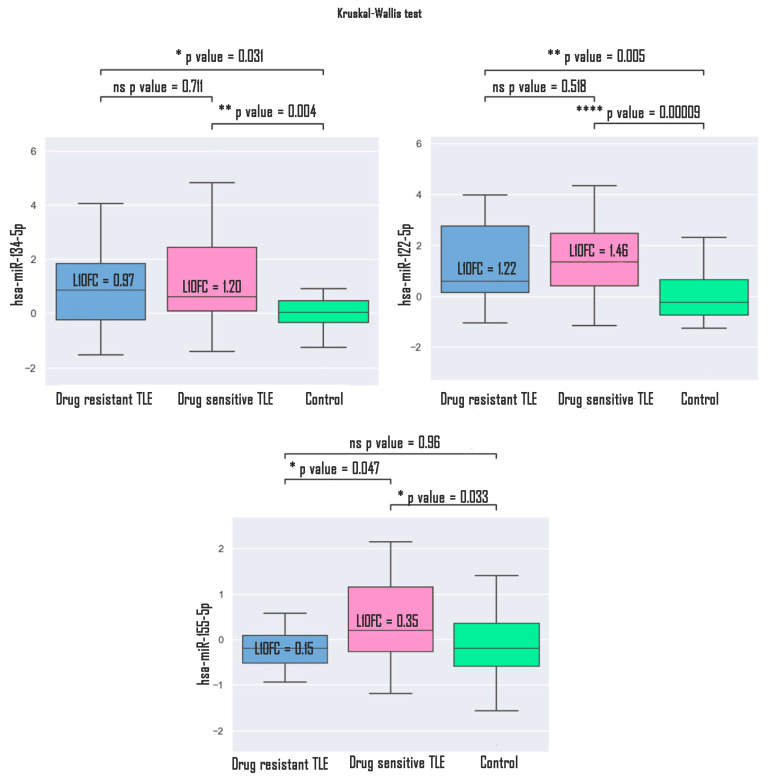
Difference in expression patterns in DR and DS TLE patients. * *p* value < 0.05.

**Figure 9 medsci-13-00007-f009:**
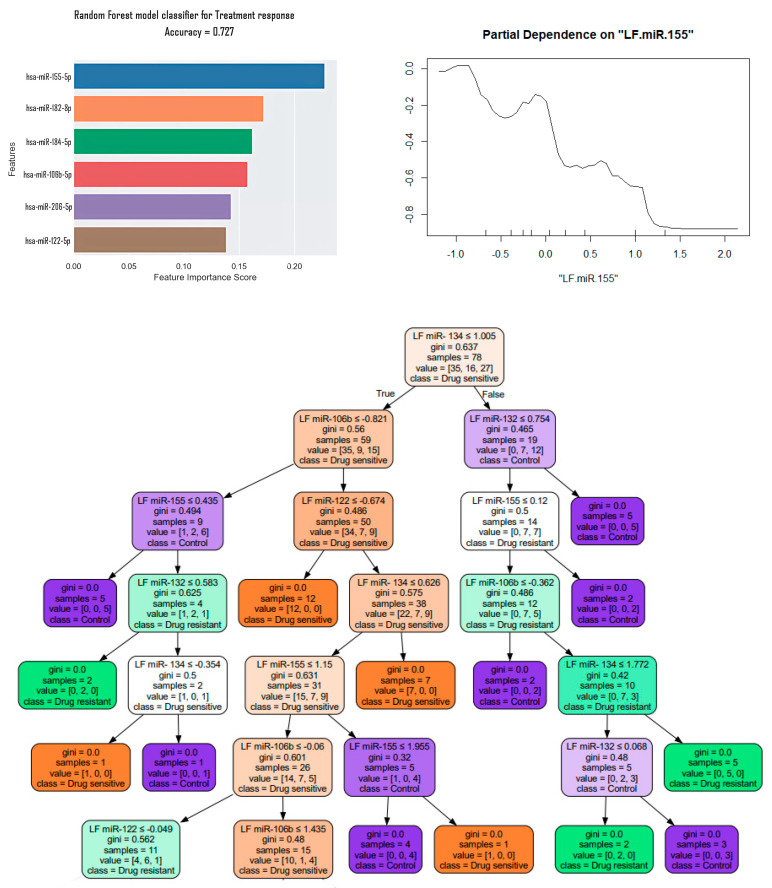
Classification models for drug-resistant and drug-sensitive group (gini-Gini index).

**Figure 10 medsci-13-00007-f010:**
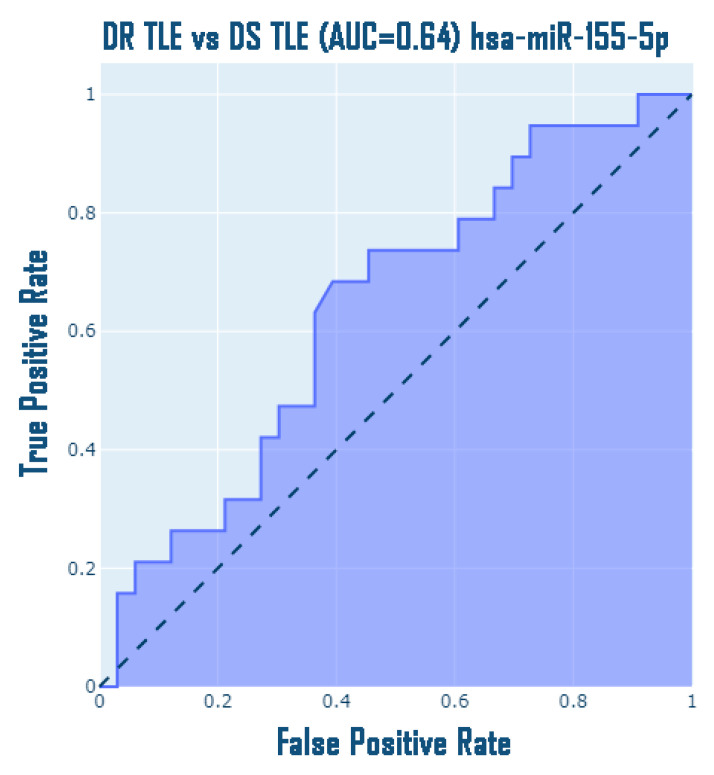
ROC curve for DR and DS patients with TLE.

**Figure 11 medsci-13-00007-f011:**
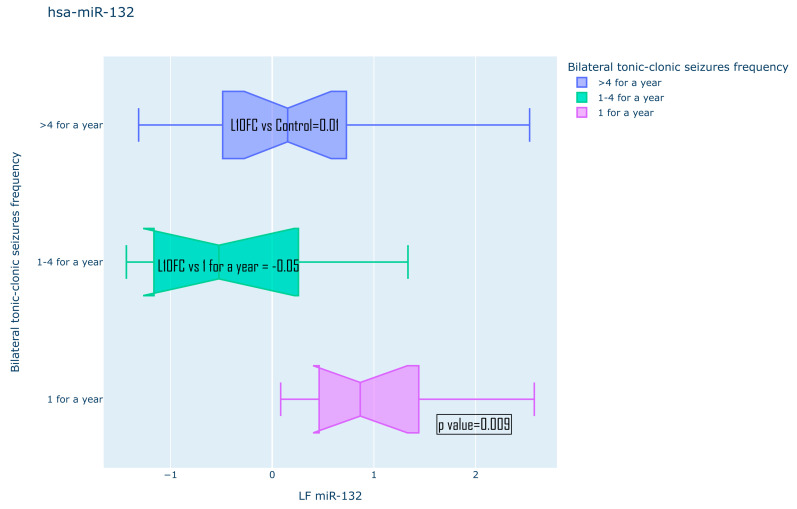
Expression pattern of has-miR-132-5p for different BTCS frequency.

**Figure 12 medsci-13-00007-f012:**
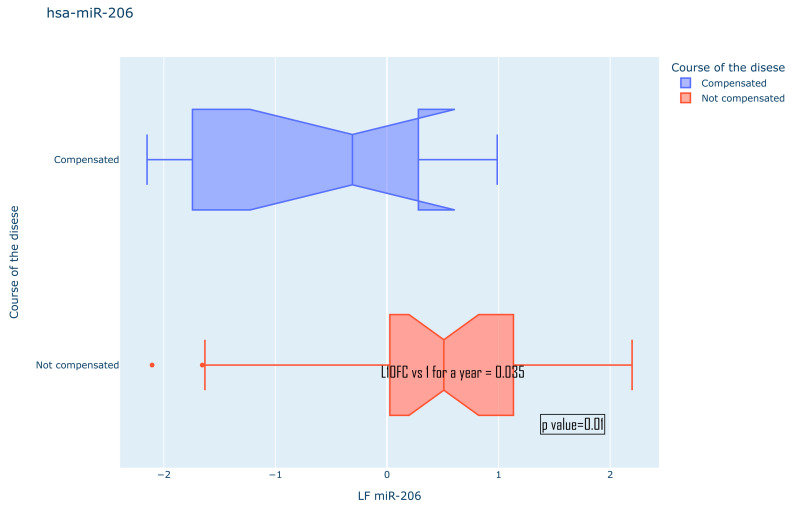
Expression pattern of has-miR-206 for different compensated and not compensated course of JME.

**Figure 13 medsci-13-00007-f013:**
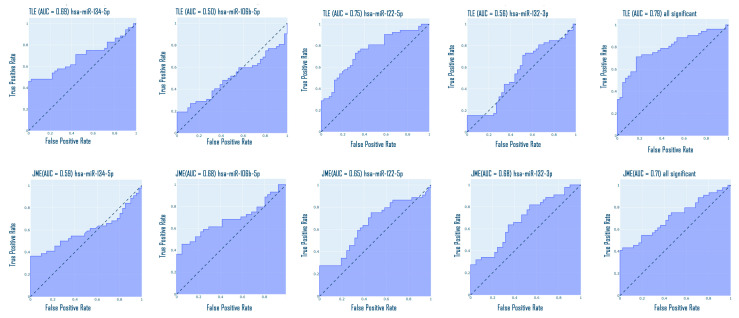
ROC curves for different epileptic syndromes.

**Figure 14 medsci-13-00007-f014:**
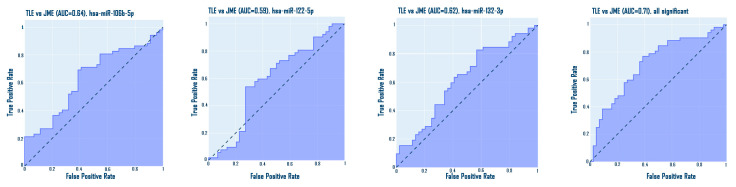
ROC curves between TLE and JME.

**Figure 15 medsci-13-00007-f015:**
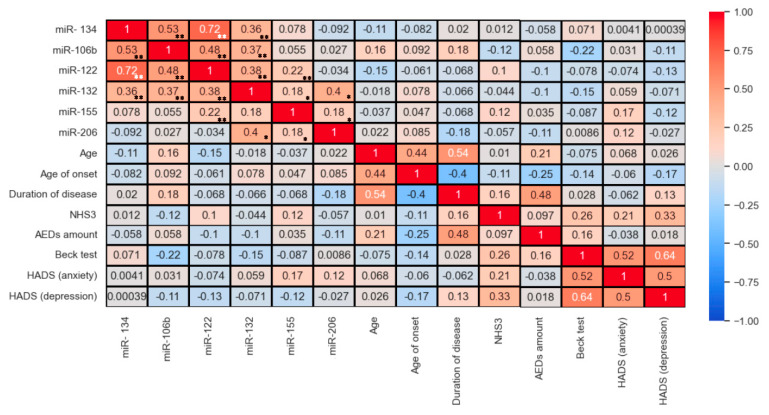
Spearman matrix correlation (* *p* < 0.05; ** *p* < 0.01).

**Figure 16 medsci-13-00007-f016:**
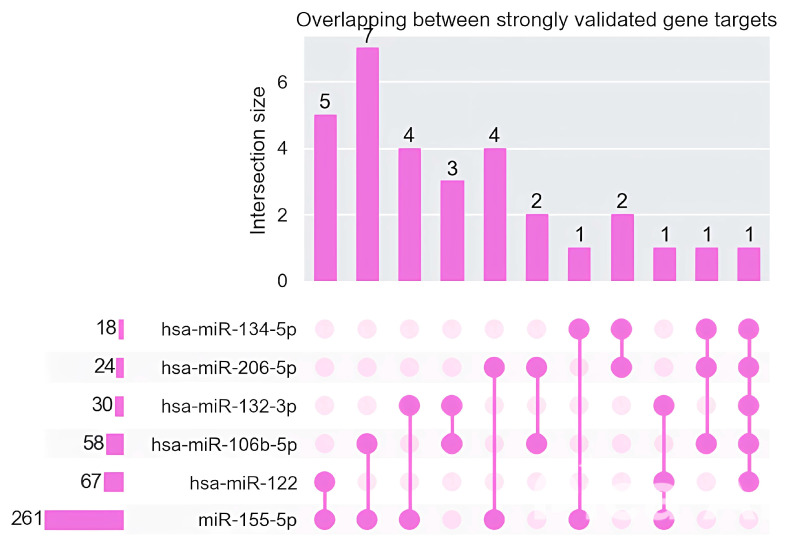
Overlapping between gene targets for studied microRNAs.

**Figure 17 medsci-13-00007-f017:**
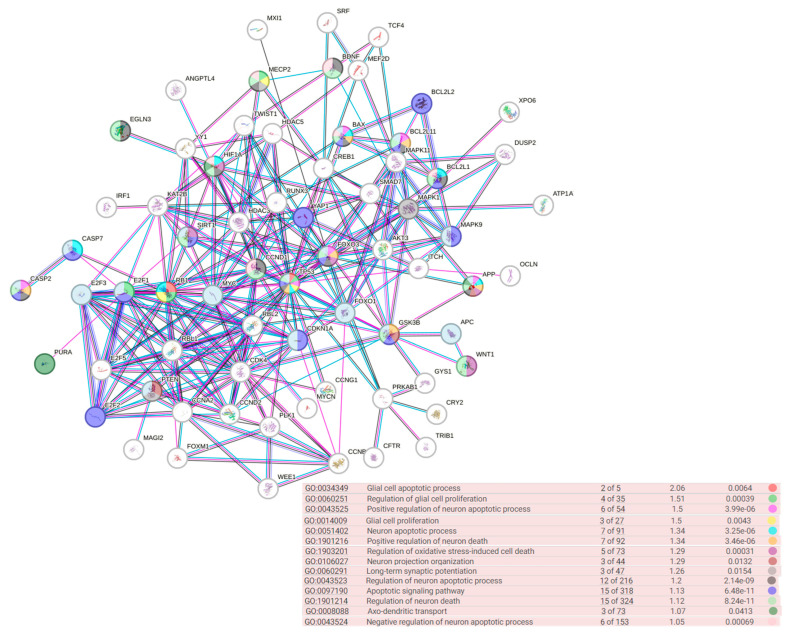
Biological processes regulated by gene targets under studied microRNAs.

**Table 1 medsci-13-00007-t001:** Patient’s characteristics.

Variable	Group			*p* Value
	TLE	JME	Control	
Gender				
Males: n, %	27 50.94%	15 35.71%	14 34.15%	χ^2^ 0.179
Females: n, %	26 49.06%	27 64.29%	27 65.85%	
Age				
Young age (<40): n, %	44 81.48%	40 95.24%	33 80.49%	χ^2^ 0.094
Middle age (>40): n, %	10 18.52%	2 4.76%	8 19.51%	
Disease duration				
>10 years	29 54.72%	18 45.00%		χ^2^ 0.509
5–10 years	14 26.42%	15 37.50%		
<5 years	10 18.87%	7 17.50%		
Age of onset	21 [15; 30]	15.5 [13; 17]		U test 0.003
Disease duration	13 [7; 21]	10 [5; 16.75]		U test 0.193
NHS-3	12 [7.25; 17.75]	16 [10; 19.5]		U test 0.411
HADS (anxiety)	5 [4; 9]	7 [5; 9]		U test 0.224
HADS (depression)	4 [2; 7]	5 [2; 9.75]		U test 0.238
AEDS amount	2 [1; 3]	2 [1; 3]		U test 0.173

NHS-3—National Hospital Seizure Severity Scale; HADS—Hospital Anxiety and Depression Scale; AEDs—Antiepileptic Drugs.

**Table 2 medsci-13-00007-t002:** Difference in expression patterns between different groups.

microRNA	Feature		*p* Value TLE	*p* Value JME
134-5p	Serial course	Not serial course	0.829	0.986
106b-5p			0.674	0.408
122-5p			0.578	0.396
132-5p			0.369	0.928
155-5p			0.169	0.102
206-5p			0.215	0.59
134-5p	Not compensated	Compensated	0.29	0.967
106b-5p			0.523	0.77
122-5p			0.218	0.897
132-5p			0.42	0.089
155-5p			0.773	0.152
206-5p			0.726	**0.0104 ***
	BTCS/GTCS (>4 for a year/1–4 for a year/1 for a year)			
134-5p			0.253	0.313
106b-5p			0.272	0.946
122-5p			0.129	0.734
132-5p			**0.009**	0.459
155-5p			0.431	0.946
206-5p			0.165	0.459

* *p* value < 0.05.

## Data Availability

The data presented in this study are available on request from the corresponding author. The data are not publicly available due to internal regulations.

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
