# Peer review of "Circulating microRNAs as Biomarkers of Various Forms of Epilepsy"

_medsci, 2025, doi:10.3390/medsci13010007_

Round 1

Reviewer 1 Report

Comments and Suggestions for Authors

Epilepsy is one type of disorder affecting health in the world. Although current clinical methods have provided proven diagnosis of epileptic syndromes. There is a great need to identify additional markers of the disease that can complement and facilitate the diagnosis of epileptic syndromes. Circulating molecular markers for epilepsy diagnosis has its advantages for epilepsy diagnosis. This manuscript studied expression profiles of 6 microRNAs including hsa-miR-106b-5p, hsa-miR-134-5p, hsa-miR-122-5p, hsa-miR-132-3p, hsa-miR-155-5p, hsa-miR- 206-5p between epileptic patients and health volunteers, finding that 4 of them show significant expression difference, therefore providing evidence that microRNAs can be as plasma biomarkers for diagnosing various epileptic syndromes. Overall, this manuscript was written and organized well. It is of interest to authors for Medical Sciences. However, I have some comments that need to be addressed before it can be accepted for publication. Below are my comments.

Major comments

1. In the methods, authors showed that patients are with big age differentiation from 18-55. Does such big age gap really provide accurate results for testing microRNA difference? We can also see obvious error bar from such as figures 3-5. 

2. Also, do authors know if individual has any microRNA expression difference at different stage from first time onset to current stage? This may affect the accuracy of diagnosis using these biomarkers if they may peak depending on stages.

3. Another key question is do authors think if it will be more accurate to compare these increased microRNAs to some more stable biomarkers from the same patient? Will such method standard decrease the result bar?

4. Authors showed so many figures in this manuscript. Do authors think it will be better to only show the most important findings? 

Minor comments

1. Several figures are difficult to read due to the font size such as figures 2,3. 

2. Statistical significance should be added in figure legends. 

Author Response

Dear reviewer!

We really appreciate the time you spent reading our manuscript and your comments. We have tried to answer any questions you may have and clarify some points. All modifications in the manuscript are highlighted in yellow.

1) Unfortunately, research work with clinical material is always associated with a number of limitations. Stricter age thresholds may make it impossible to recruit a group for subsequent analysis. However, the median age of people recruited for the study was 30 [25.00; 38.75]. It was for the purpose of eliminating the influence of age on the expression of microRNAs that we performed an additional analysis of correlations, including age and expression of selected microRNAs using Spearman's method. In addition, we separately noted the fact of a wide range of ages as a limitation.

2) All patients recruited for the study were in the chronic phase, this point was not additionally noted in the text, but Table 1 shows the median duration of the disease. Studying the expression of microRNAs at different stages of the disease is a really important and urgent task, which will be the subject of our further study using a disease model, because Using the model allows you to more clearly determine the stages of the disease and obtain biomaterial.

3) To date, we have published two papers on protein plasma markers of temporal lobe epilepsy: 10.3390/metabo13010083; 10.3390/ijms25147935, the latter of which used a subset of the sample from our current study. In the future, we plan to expand the sample under study and also conduct an analysis of protein expression in patients with JME, which will allow us to fully analyze the correlation between the expression of microRNAs and proteins that may be the products of potential gene targets. In this work, we limited ourselves to the expression of microRNAs.

4) The large number of figures ensures complete transparency of all the tests we have carried out.

The font of the figures and the figures themselves have been upscaled. Each figure has the corresponding p value, but it may not have been visible because the font was too small.

Reviewer 2 Report

Comments and Suggestions for Authors

-           Please try to re-do the abstract without repeating so many times the word “Epilepsy” and avoid the use of abbreviations (e.g. TLE, JME) A better fluidity of the text increases the interests of potential readers.

-           The definition that authors present according to ILEA does not refer to “epilepsy” but rather to “epilepsy syndromes”. Please review.

-           Please review and improve grammar throughout all manuscript;

-           On introduction, the authors state that “It is known that many microRNAs are specifically expressed in different cells of the brain [22], making them good candidates for epilepsy biomarkers”. However, many cell types can be involved in ictogenesis. Therefore, explain how this can be an advantage.

-           Why choose TLE and JME and not others (or more) to robust the study?

-           Please explain why there was a need to perform HADS and NHS3 tests to the patients if the diagnosis was previously performed by a trained neurologist.

-           Review “incl.” on line 101. “Anamnesis” paragraph should also be clearer.

-           According to literature research for selecting the microRNAs intended to be analyzed, were there additional ones that were not considered by the authors in this study? Were the six selected specifically reported for any type of epileptic syndrome in the analyzed studies found in literature?

-           “Enrichment of the brain with the studied microRNAs allows for further analysis.” This sentence is not clear (line 161);

-           Authors should consider reviewing figure 1 and make it simpler. There is also one grey dot without identification;

-           Figures have very poor quality to be published;

-           Was there a tendency for the biomarker’s findings regarding each type of antiseizure therapy of patients? Or the authors did not record each patient therapy?

-           Which methods do authors consider to be machine learning methods?

-           On Figure 7 and 9, what is “gini”? Please improve and complete all figures captions including the meaning of abbreviations and the statistical tests applied (when applicable);

-           This study, even though being very interesting, can be a very simplistic approach to distinguish the studied microRNAs as specific biomarkers for the 2 epilepsy syndromes studied. There should also be considered other epileptic syndromes, cognition and lifestyle of patients (exercise practice, nutrition, daily activity), scientific methods limitations (stability of sample; selectivity and sensibility of applied tests), proteins expression analysis, etc. The authors should discuss the study limitation more deeply and possible future perspectives using the collected data.

Author Response

Dear Reviewer!

Thank you very much for your time and comments. We have tried to answer your questions and made modifications to the body of the manuscript. All modifications are highlighted in yellow.

  • Thank you for these comments, the word “epilepsy” actually occurs quite often, we tried to correct this, abbreviations were also removed from the abstract.
  • Modified this in the text of the manuscript.
  • We tried to correct all grammatical mistakes in the text.
  • Perhaps we did not understand your question quite correctly. Ictogenesis, according to the definition, corresponds to neuroglial mechanisms that lead to spontaneous or reflexive outbreaks of epileptic activity in the brain. Indeed, both neuronal and glial cells can participate in ictogenesis. Therefore, we did not limit this proposal to, for example, neuronal cells of the brain. In this context, "expressed in the brain" is not a limitation, since ictogenic events, which can be accompanied by aberrant expression of both transcripts and protein, occur in the brain.
  • We chose two different forms of epilepsy: with focal seizures, as well as a genetic form due to the significant differences between them in seizure types, etiology, etc. The choice of TLE and JME is due to the fact that these are the most common forms of epilepsy, which are generally accepted models for studying epilepsy. In addition, patients with JME also describe microdysgenesis of the brain, which allows us to compare these two groups of patients. The purpose of the work was to study the expression of microRNA in very different types of epileptic disorders, using both of them as comparison groups. Also, answering this question, why TLE and JME: at the university clinic, where the biomaterial was collected, the largest number of patients were represented by these diagnoses, which made it possible to collect samples of comparable size. Inclusion of other forms of genetic epilepsy in the study, such as, for example, childhood absence epilepsy or childhood myoclonic epilepsy, would significantly affect the age characteristics of the patients, imposing additional limitations on the study. To date, we are conducting a study of childhood genetic forms of epilepsy in the context of microRNA expression; upon its completion, it will be possible to conduct additional analysis with known amendments and limitations.
  • Our study was conducted in the neurological center of the University Hospital, specializing in the diagnosis and treatment of patients with epilepsy, the assessment of the HADS NH3 scales was carried out by specially trained neurologists. We additionally introduced the HADS and NHS3 scales to be able to analyze the correlations between the scores obtained on these scales and expression.
  • We have modified the paragraph with the anamnesis.
  • We selected only six microRNAs for our study. Yes, there is another pool of microRNAs described in the literature that we did not consider in our study, in particular, there are extensive reviews of microRNA expression in epilepsy, such as: 10.3389/fnmol.2021.650372. No, not all microRNAs we studied have been registered for all types of epilepsy, in particular: hsa-miR-122 has not been studied in cases of epilepsy, but is considered to be associated with neuroinflammation (10.1007/s11064-023-04014-7, 10.1186/s12974-024-03162-z), as well as the induction of neuronal cell apoptosis (10.1016/j.cellsig.2023.110668). The remaining microRNAs studied were mostly studied in cases of temporal lobe epilepsy, while studies of genetic forms are very limited.
  • Since most studies of microRNA expression in epilepsy, conducted on clinical material, studied biological fluids, it was important for us to establish that all of the microRNAs included in our study are detected in the brain, which is the main pathological substrate of this disease. In this context, “Enrichment of the brain” implies that the microRNAs we selected are detected and widely expressed in the brain.
  • We have added a simplified version of Figure 1, which presents a heatmap of the expression of the studied microRNAs in different regions and cells of the brain. A non-zero logarithm demonstrates the detection of microRNAs in a particular region.
  • The quality of the figures has been improved.
  • We did not record each type of therapy for patients, nor did we divide patients into groups depending on the type of antiepileptic therapy.
  • We used machine learning algorithms, the word "methods" is not quite correct and it was changed in the body of the manuscript. Among the algorithms used: Random Forest Classifier and Decision Tree. Both algorithms were trained on a test sample, which made up 20% of the entire sample.
  • We have added figure captions and additional information in the body of the manuscript about this parameter. Gini Index, also known as Gini impurity, calculates the amount of probability of a specific feature that is classified incorrectly when selected randomly. If all the elements are linked with a single class then it can be called pure.
  • Indeed, in our work we considered only two epileptic syndromes, while there are a significant number of them, which limits our study, we supplemented our conclusion with this limitation. In this work, we did not focus on the cognitive abilities of patients, as well as the daily routine included in the study, continuation and deepening of this topic in this area can really significantly expand the understanding of the role of microRNA in pathogenesis, thank you for this idea. As for the study of other markers, to date we have published two papers devoted to protein plasma markers of temporal lobe epilepsy: 10.3390 / metabo13010083; 10.3390 / ijms25147935, the latter of which used some part of the sample from our current study. In the future, we plan to expand the studied sample, as well as conduct an analysis of protein expression in patients with JME, which will allow a full analysis of the correlation of microRNA expression and proteins that may be products of potential gene targets. In this work, we limited ourselves to the expression of microRNAs.

Best regards, authors.

Reviewer 3 Report

Comments and Suggestions for Authors

This paper reflects in every part and point the characteristics of an accurate description of pathology and possible variants, paying special attention to two forms of epilepsy, TLE and JME, appropriately compared with a control.

The idea of evaluating the expression of various micro-RNAs by plasma obtained from simple blood sampling allows for easy reproducibility and most importantly non-invasiveness of the clinical approach, which is of paramount importance in the study of pathologies.

The number of patients recruited makes it possible to identify an important clinical picture, such that the veracity of the data is guaranteed.

The images included in the work are clear and specific in each point treated.

I have just one question for the authors, or rather a pure curiosity, have they thought of assessing micro-RNA levels from lymphocytes or more generally from PBMCs rather than plasma? If not, it could be considered for further study to see if the inflammatory process may be accompanied by changes in such cells.

Minor Review:

Line 348: It is likely to miss the closing sentence point before “this”.

Author Response

Dear Reviewer!

Thank you for your time and comments! Thank you for this suggestion, it is a really interesting idea that we can implement, as our biobank contains blood samples from patients included in this study. Line 348: modified.

Best regards, authors.

Round 2

Reviewer 1 Report

Comments and Suggestions for Authors

Thank authors for addressing my concerns. I have no more comments, recommending acceptance for publication.

Author Response

Dear reviewer!

Thank you again for your feedback and recomendance for acceptance.

Best regards, authors.

Reviewer 2 Report

Comments and Suggestions for Authors

The visual quality of the figures needs to be improved. Try changing the type of figures to TIFF 300 dpi or using other software to improve them.

The authors have answered all the questions, although I still have some doubts about the novelty of this study (for example, as reported by the authors, some of the microRNAs studied have already been reported in other scientific articles). Even so, I think it is of sufficient quality to be published in the journal Medical Sciences.

Author Response

Dear reviewer!

We have attached our response as a separate file.

Best regards, authors.
